# Beyond the ABCs—Discovery of Three New Plasmid Types in *Rhodobacterales* (RepQ, RepY, RepW)

**DOI:** 10.3390/microorganisms10040738

**Published:** 2022-03-29

**Authors:** Heike M. Freese, Victoria Ringel, Jörg Overmann, Jörn Petersen

**Affiliations:** 1Leibniz-Institut DSMZ—Deutsche Sammlung von Mikroorganismen und Zellkulturen GmbH, Inhoffenstraße 7 B, 38124 Braunschweig, Germany; victoria.ringel@dsmz.de (V.R.); joerg.overmann@dsmz.de (J.O.); 2Institut für Mikrobiologie, Technische Universität Braunschweig, Spielmannstraße 7, 38106 Braunschweig, Germany

**Keywords:** cryptic plasmids, replicase, plasmid classification, chromids, evolution, phylogenomics

## Abstract

Copiotrophic marine bacteria of the Roseobacter group (*Rhodobacterales*, *Alphaproteobacteria*) are characterized by a multipartite genome organization. We sequenced the genomes *of Sulfitobacter indolifex* DSM 14862^T^ and four related plasmid-rich isolates in order to investigate the composition, distribution, and evolution of their extrachromosomal replicons (ECRs). A combination of long-read PacBio and short-read Illumina sequencing was required to establish complete closed genomes that comprised up to twelve ECRs. The ECRs were differentiated in stably evolving chromids and genuine plasmids. Among the chromids, a diagnostic RepABC-8 replicon was detected in four *Sulfitobacter* species that likely reflects an evolutionary innovation that originated in their common ancestor. Classification of the ECRs showed that the most abundant plasmid system is RepABC, followed by RepA, DnaA-like, and RepB. However, the strains also contained three novel plasmid types that were designated RepQ, RepY, and RepW. We confirmed the functionality of their replicases, investigated the genetic inventory of the mostly cryptic plasmids, and retraced their evolutionary origin. Remarkably, the RepY plasmid of *S. pontiacus* DSM 110277 is the first high copy-number plasmid discovered in *Rhodobacterales*.

## 1. Introduction

Bacteria are capable of rapidly adapting to changing environments via the acquisition of mobile genetic elements [1,2,3]. Proteobacterial plasmids carry a wide range of accessory genes that are beneficial for their bacterial hosts and might even pave the way for the colonization of novel environmental niches. In addition to the spread of antibiotic resistances [4,5,6], they provide protection against heavy metal and xenobiotic pollutants [3,7,8], facilitate interactions with algae and plants [9,10], mediate the formation of biofilms [11], and allow the utilization of specific carbon sources [12,13]. Even large genetic units with sizes of more than 40 kb, such as the gene clusters for aerobic anoxygenic photosynthesis or the formation of functional flagella, are occasionally encoded on extrachromosomal replicons (ECRs) [14,15].

Regardless of the highly variable accessory genes, plasmids are characterized by their essential replicase, and a set of conserved backbone genes required for their maintenance and transfer [3,16]. Low copy number plasmids comprise a partitioning system for stable maintenance homologous to those of the bacterial chromosome. It consists of the centromere-like binding site, a DNA-binding protein (ParB), and a motor protein (ParA) that mediates a concerted transfer of the replicated plasmids to the cell poles, thus ensuring their reliable distribution to the daughter cells [17]. In contrast, high copy number plasmids lack a partitioning system and are randomly distributed during bacterial cell division. Other characteristic backbone genes are toxin-antitoxin systems representing addiction modules that prevent the spontaneous loss of a plasmid [18]. The horizontal transfer of plasmids is essentially mediated by conserved type IV secretion systems (T4SSs) that encode the crucial relaxase for the release of single-stranded DNA, a coupling protein, and a sophisticated nanotube for their conjugative transmission [19]. Many cryptic plasmids, which are small selfish replicons without beneficial genes for the host, contain only mobilization (MOB) genes encoding the pivotal relaxase and therefore require conjugative plasmids with T4SSs for their interbacterial exchange [20]. The majority of plasmids lack the known MOB genes and are therefore considered non-mobilizable, although comparative analyses suggested that they were once horizontally transferred [20,21].

The only indispensable unit of a plasmid is its replication system, which includes the origin of replication (*oriV*) and a diagnostic replicase. Replication proteins are reliable markers for the plasmid classification, and phylogenetic analyses of homologous replicases even allow the discrimination of different compatibility groups [22,23]. A trustworthy and reproducible classification of ECRs is crucial for the comparison of plasmids from different bacterial lineages and the identification of novel plasmid types. Based on the phylogenetic approach, six plasmid types were so far identified within the alphaproteobacterial order *Rhodobacterales*. These are RepA, RepB, RepABC, DnaA-like, RepL, and RepC_soli plasmids that encompass up to nine different compatibility groups [4,7,24]. However, an analysis of small plasmids from four *Paracoccus* strains indicated the presence of further replicon types [25].

Many of the mentioned plasmid types were first detected in roseobacters (*Roseobacteraceae*), a highly metabolically and ecologically versatile group of marine *Rhodobacterales* [26,27]. Copiotrophic roseobacters typically exhibit a multipartite genome organization and are thus particularly suitable models to investigate the diversity and biology of plasmids [7,15]. However, a prerequisite for a systematic assessment of bacterial plasmids and the detection of novel replicon types is the availability of complete closed genomes in which all extrachromosomal elements were sequenced, e.g., [28]. Although more than 3000 *Rhodobacterales* genomes are available at the NCBI, over 90% of them are draft versions, which prevent the detection and unambiguous assignment of uncharacterized replicases to ECRs and thus the identification of novel plasmid types. In the current study, we established closed genomes of five different roseobacter strains from the genera *Sulfitobacter* and *Pseudosulfitobacter*, which was recently separated from *Sulfitobacter* as a novel genus [29]. These genera contain metabolically versatile and biogeographically widespread bacterial generalists [30,31,32]. Comparative analyses revealed the presence of three novel plasmid types designated RepQ, RepY, and RepW.

## 2. Materials and Methods

### 2.1. Bacterial Strains and Growth Conditions

Five bacterial *Sulfitobacter* strains (DSM 14862, 2RS2_G6, 3RS2_G4b, SO248Ex84, PIC-76) were investigated in this study. Strain DSM 14862^T^ was obtained from the Leibniz Institute DSMZ. Strains 2RS2_G6 and 3RS2_G4b were isolated from sediment samples collected in the Channel Sea near Roscoff (48.7205, −3.9651) on the 26 September 2013. Strain SO248Ex84 was isolated by Sara Billerbeck (Institute for Chemistry and Biology of the Marine Environment [ICBM], Oldenburg, Germany) from Pacific Ocean seawater collected during the Sonne cruise SO248 (45.0001, 178.7499) on the 24 May 2016. Strain PIC-76 was provided by Irene Wagner-Döbler [33]. All strains are now deposited at the Leibniz Institute DSMZ under the following numbers: DSM 14862^T^, DSM 110093, DSM 109990, DSM 110277, and DSM 107133.

Functionality tests of putative replication genes were conducted with *Phaeobacter inhibens* DSM 17395 Δ65/3 lacking the biofilm chromid for surface attachment [4]. *P. inhibens* and the (*Pseudo*)*sulfitobacter* strains were cultivated in marine broth medium (MB, Carl Roth, or Difco) at 28 °C and 20 °C, respectively. Competent NEB^®^Turbo *Escherichia coli* cells (New England Biolab), used for cloning, were cultivated in Luria-Bertani medium (LB, Carl Roth) at 37 °C. The selection of transformants was performed on 0.5× MB or LB plates with 120 µg mL^−1^ kanamycin (KAN).

### 2.2. Genome Sequencing, Assembly and Annotation

Strains were harvested in the early stationary phase. Genomic DNA was extracted with the QIAGEN Genomic-tips 100 Kit (Qiagen, Hilden, Germany). SMRTbell™ template libraries were prepared according to the recommended instructions, 5 µg of genomic DNA were end-repaired and ligated to hairpin adapters using P6 chemistry (Pacific Biosciences, Menlo Park, CA, USA). BluePippin™ Size-Selection to greater than 3 kb was performed according to the manufacturer’s instructions (Sage Science, Beverly, MA, USA). SMRT sequencing was carried out on the PacBio RSII platform (Pacific Biosciences). To evaluate the recovery of small ECRs, deep SMRT sequencing of libraries that were constructed without size selection was performed on the Sequel II system (Pacific Biosciences), taking a 15 or 30 h movie. Illumina libraries were prepared with the Nextera XT DNA Sample Preparation Kit (lllumina Inc., San Diego, CA, USA) with a modified protocol [34], and paired-end Illumina sequencing was either performed on the NextSeq 500 (PE75 or PE150) or for strain PIC-76 on the MiSeq platform (PE300).

PacBio reads were assembled de novo with HGAP3 in SMRT Portal 2.3.0 or for the exemplary Sequel reads using the “Microbial Assembly” protocol in SMRT Link 10.0.0. The assembled contigs were error-corrected by mapping of Illumina short reads using the Burrows-Wheeler Aligner (BWA 0.6.2) [35] and subsequent variant and consensus calling using VarScan 2.3.6 [36]. An additional Illumina short read assembly was performed using velvet 1.2.10 [37], which was also corrected as mentioned before. Redundancies in parallel assemblies were identified with the Genome Finishing Tool of the CLC Genomics Workbench 7.0.1 and removed. The resulting assembly was trimmed, circularized, and adjusted to the replication system as a start point (https://github.com/boykebunk/genomefinish, accessed on 1 February 2022), checked via mapping of Illumina (BWA) and PacBio reads (RSII: Bridgemapper; Sequel: Resequencing) and finally corrected if necessary. Methylation motifs were detected with the RS Modification and Motif Analysis in SMRT Portal. The genomes were annotated with Prokka 1.13 [38] with a subsequent manual curation of the replication systems. The complete genomes are deposited in NCBI GenBank under the accession numbers: CP084951-58, CP084959-65, CP085144-53, CP085154-66, and CP085167-72.

### 2.3. Characterization and Comparison of Genomes, Phylogenetic Analyses and Data Handling

Illumina reads were mapped on the corresponding final assembly with BWA 0.6.2 [35] to estimate the copy number of the ECRs. The median coverage per element was calculated from the coverage per position extracted with SAMtools (v0.1.19) [39]. The copy number of ECRs was calculated as the ratio of their median coverage to the median chromosomal coverage. In order to differentiate genuine plasmids from chromids [40], four criteria were applied: (i) All chromids contained a *parAB* partitioning operon, (ii) represented low copy number replicons with coverage below two, (iii) had a GC content that deviated less than 2.5% from the chromosome, and (iv) showed a genomic imprint that was comparable to those of the chromosome. The last criterion was independently investigated with clustering analyses of the relative synonymous codon usage (CU) and the tetranucleotide frequency (tetra), both performed in R version 4.1. The well-characterized genome of *Dinoroseobacter shibae* DFL12 (GCA_000018145.1) served as an internal reference [41]. However, ECRs were only classified as chromids if the results of both methods (CU, tetra) agreed.

All replicons were screened for the presence of a type IV secretion system (T4SS) or a characteristic relaxase (*mob* with MOBscan https://castillo.dicom.unican.es/mobscan/, accessed on 29 September 2021), which is required for plasmid mobilization [20,42].

Taxonomic classification of the five investigated strains were based on genome sequences of all type strains from *Sulfitobacter*, *Roseobacter,* and *Pseudosulfitobacter* (including non-validated ones) mentioned at LPSN [43]. Further genomic comparisons were made with all closed *Sulfitobacter* genomes available at NCBI in October 2021, as well as closed genomes of 24 reference strains representing different clades within the *Rhodobacterales* (see Appendix A). A genome-based phylogenetic tree was reconstructed from a concatenated amino acid alignment of 92 housekeeping genes generated with UBCG [44]. An approximately Maximum Likelihood phylogenetic tree was inferred using FastTree version 2.1.11 (double-precision) with default parameters [45]. The digital DNA-DNA hybridizations of selected strains were calculated with TYGS [46]. Replication systems of ECRs were identified and classified [4,7,24,47].

Novel replicases were compared with the Pfam database [48]. RepQ, RepY, and RepW type plasmid replication proteins have further been identified with BLASTP searches in the NCBI database. Most replicases were located on contigs from draft genomes, which makes it difficult to draw clear conclusions about their extrachromosomal localization. Accordingly, the 5′ and 3′ end of all linear contigs were compared via BLASTN, and a replicon was defined as circular if at least 50 nucleotides overlap with 100% identity.

The protein sequences of new replicases (RepQ, RepY, RepW) were aligned with muscle [49], and all positions containing gaps and missing data were eliminated. The evolutionary history was inferred by the Maximum Likelihood method based on the JTT matrix-based model [50]. The analyses, with 100 bootstrap replicates, were conducted in MEGA7 [51]. Initial tree(s) for the heuristic search were obtained automatically by applying Neighbor-Join and BioNJ algorithms to a pairwise distance matrix estimated using a JTT model, and then selecting the topology with the best log likelihood value.

Color and font style of these trees were modified in PowerPoint, the other figures were generated in R 4.1 with the packages ggplot2 [52], ggtree [53] and gggene [54].

### 2.4. Functionality Test of RepQ, RepY and RepW

Plasmid replication systems of the five ECRs with new replication systems were amplified with the Phusion^®^ High-Fidelity DNA Polymerase, PCR products were cloned into the commercial vector pCR2.1, and the absence of PCR errors was confirmed via Sanger sequencing as previously reported [4]. Primer sequences and the respective size of the PCR products are listed below: (i) RepQ module of the *Sulfitobacter indolifex* DSM 14862^T^ plasmid pDSM14862_g (P2211: 5′-AGGGTATGGCGACGGTAAAC-3′, P2210: 5′-TAGGGATGGTAGGAGTGGAGAG-3′; 2579 bp), (ii) RepY module of the *S. pontiacus* DSM 110277 plasmid pDSM110277_f (P2193: 5′-AGCACCAAGCAAGATGATAC-3′, P2194: 5′-GAGCGAGACACCCTTTTTAC-3′; 3423 bp), (iii) RepW module of the *S. dubius* DSM 109990 plasmid pDSM109990_i (P2206: 5′-GAGGGAAGGGGAGAAGAAAC-3′, P2207: 5′-CATCAACAGCCACAGGATAC-3′; 2350 bp), (iv) RepW module of the *Sulfitobacter* sp. DSM 110093 plasmid pDSM110093_e (P2208: 5′-TTTGCTCATACCGTTTCTCC-3′, P2209: 5′-CTTTGTTCGCCGTTTTACTC-3′; 2492 bp), and (v) RepW module of the *Pseudosulfitobacter* sp. DSM 107133 plasmid pDSM107133_l (P2547: 5′-ACGGTGAGTCCCAGATCAGA-3′, P2546: 5′-GCTTTGTTCGGTTTGACGCT-3′; 1992 bp). The functionality of the replicases was tested in the plasmid-cured *P. inhibens* DSM 17395 Δ65/3 mutant lacking the 65 kb biofilm chromid as previously described [4,55]. The standardized test used the re-isolation of the plasmid construct and its transformation in *E. coli* to exclude chromosomal integration of the construct, including the kanamycin resistance marker. This procedure provides the final proof for active replication of the construct in *P. inhibens*.

## 3. Results and Discussion

### 3.1. Genome Sequencing and Classification of Five (Pseudo)Sulfitobacter Strains

To generate closed and complete genomes of the (*Pseudo*)*sulfitobacter* strains, at least two PacBio HGAP3 assemblies were computed. A reproducible outcome irrespective of the applied assembly parameters, such as the specified genome size, served as an internal control for the reliability of the assembly. Additionally, an Illumina velvet assembly was conducted, and all assemblies were compared to retrieve non-redundant contigs. Both assembly strategies were necessary because smaller plasmids (<10 kb) were often not assembled with PacBio reads due to shearing of the DNA and the size selection of the libraries. For instance, the PacBio assembly of *S. dubius* DSM 109990 only comprised the chromosome and five extrachromosomal replicons with sizes between 48 and 284 kb (Table 1). Four smaller plasmids ranging from 3.8 to 6.3 kb in size would have been overlooked without the additional Illumina assembly. Subsequent mapping of PacBio sequences on these plasmids revealed a very low number of long reads (Appendix A in comparison to Appendix A), which were insufficient for a *de novo* assembly, but allowed to confirm the results. On the contrary, the mapping of the Illumina reads even showed that those small plasmids had 10 to 15 times higher copy numbers than the chromosome (Table 1). As both sequencing libraries were prepared from the same DNA extract, small plasmids were obviously quantitatively depleted during the PacBio library preparation. Even high-throughput sequencing (>5000x coverage) without size selection on the novel PacBio Sequel system could not compensate for this effect. The assemblies of medium-sized plasmids also suffered from that depletion. For instance, the 17 kb replicon pDSM14862_g of *S. indolifex* with 5.3 copies per cell reached only 3% of the chromosomal PacBio coverage (pDSM14862_g: 13x, chromosome: 434x). Altogether, ten plasmids would have been overseen in five strains investigated in the current study if only long-read assemblies had been performed. We conclude that the establishment of high-quality complete bacterial genomes requires a combination of long- and short-read sequencing combined with manual data curation.

All five investigated strains belong to clade 2 of the *Rhodobacterales* (Figure 1) [56]. Strain DSM 14862^T^ is the type strain of *Sulfitobacter indolifex,* while the others represent taxonomically uncharacterized isolates. An additional well-supported phylogenomic tree with all related type strains allowed us to identify the closest relative(s) of our four new strains (Appendix A). One *Sulfitobacter* strain could be classified as *S. dubius* DSM 109990 (Reference: *S. dubius* DSM 16472^T^; digital DNA-DNA hybridization, dDDH; [57]: 70.8%) and one as *S. pontiacus* DSM 110277 (Reference: *S. pontiacus* DSM 10014^T^; dDDH: 77.1%). In contrast, the diagnostic values between strain DSM 110093 and the closest related type strain *S. dubius* DSM 16472^T^ (dDDH: 50.3%) clearly document that this strain represents a novel species. An analogous conclusion can be drawn for *Pseudosulfitobacter* sp. DSM 107133 (PIC-76), which forms a distinct branch with *Pseudosulfitobacter pseudonitzschiae* DSM 26824^T^ (Appendix A), but exhibits a low dDDH value of only 32.3%.

### 3.2. Characterization of Extrachromosomal Replicons

All five investigated strains comprise multipartite genomes harboring between five and twelve ECRs (Table 1), which is clearly above the average number of about three ECRs that was previously reported for *Rhodobacterales* ([26] Suppl. 2). *Pseudosulfitobacter* sp. DSM 107133 carries 12 ECRs and therefore represents together with “*Candidatus* Marinibacterium anthonyi” La 6, the current record holder in terms of multi-partitioning among roseobacters followed by *Marinovum algicola* DG898 with eleven ECRs [7,15]. In contrast to La 6, which harbors a large chromosome (5.6 Mbp), all investigated (*Pseudo*)*sulfitobacter* strains have chromosomes with a moderate size ranging from 3.0 to 3.6 Mbp. However, the numerous ECRs of *Pseudosulfitobacter* sp. DSM 107133, especially the 571 kb replicon pDSM107133_a, increased its genome size to 5.2 Mbp, while the genomes of the *Sulfitobacter* strains have only an average size of about 4 Mbp, which is characteristic for many roseobacters [58]. ECRs above 500 kb are commonly found in *Rhizobiales* [9,59] but were only sporadically detected in *Rhodobacterales,* i.e., in the genera *Paracoccus*, *Ruegeria,* and now *Pseudosulfitobacter* [12,60]. Although these ECR are often referred to as megaplasmids, the comparative analysis of the *Pseudosulfitobacter* sp. DSM 107133 replicons clearly showed that pDSM107133_a has a chromosome-like genetic imprint (Appendix A). Analogous to the 750 kb replicon pAMV3 of *P. aminovorans* JCM 7685, it can be classified as a chromid that is supposed to carry essential core genes [12,40,41].

The five investigated *Sulfitobacter* and *Pseudosulfitobacter* strains contain between one and three chromids (Table 1). Homologs of the sole chromid pDSM14862_c from *S. indolifex* (CP084954) are also present in the closely related strains *S. profundi* D7 (CP020695), *Sulfitobacter* sp. DSM 110093 (CP085170) and *S. dubius* DSM 109990 (CP085146; Table 1, Figure 1). The four chromids with sizes between 201 and 268 kb share the same RepABC-8 type replication system and exhibit a long-range synteny (Appendix A), which unequivocally documents their common origin. The absence of syntenic homologs in all other strains of Figure 1 supports the following scenario for their origin: (i) A RepABC-8 plasmid was acquired from the common ancestor of *S. dubius*, *Sulfitobacter* sp. DSM 110093, *S. profundi,* and *S. indolifex*, (ii) it stably co-evolved with the chromosome due to an essential function and (iii) was finally ameliorated into a chromid. Accordingly, this replicon exemplifies the ancient recruitment of a chromid amid the speciation process in the genus *Sulfitobacter*. Its acquisition antedates, and possibly even triggered the origin of a distinct phylogenetic lineage comprising at least four species (Figure 1).

All chromids of the investigated strains lack genes for their conjugative transfer, which is in agreement with their stable co-evolution with the chromosome. In contrast, mobilization genes are abundant on plasmids and were identified on eight of nine plasmids from *Pseudosulfitobacter* sp. DSM 107133. Four larger plasmids (>100 kb) of DSM 107133 encode T4SSs and should hence be self-transmissible via conjugation (Table 1), while pDSM107133_d and three cryptic plasmids are probably mobilizable due to the presence of relaxase-encoding MOB genes [20]. The four *Sulfitobacter* strains harbor five additional plasmids with T4SSs and six with MOB genes. The multipartite genome organization and great abundance of mobilization genes in the five investigated strains support the idea of a huge pan-mobilome of roseobacters serving as a genetic backup in a changing marine environment [61]. It likely also reflects the active role of the genera *Sulfitobacter* and *Pseudosulfitobacter* in the network of plasmid exchange in the ocean.

### 3.3. Identification of Novel Plasmid-Types (RepQ, RepY, RepW)

The ECRs of the newly sequenced (*Pseudo*)*sulfitobacter* strains encompass the replicases of all known plasmid types (RepA, RepB, RepABC, DnaA-like, RepL) described for roseobacters so far except RepC_soli (Figure 1) [4,7,24]. However, the smallest plasmid of each of the five strains lacks a replication gene that is homologous to these replicases. Comparative genome analyses suggested that the respective plasmids encode three uncharacterized replicases representing novel plasmid types. (i) RepQ: The putative replicase of the 17 kb plasmid pDSM14862_g from *S. indolifex* DSM 14862^T^ was designated RepQ in reminiscence of homologous replicases on IncQ-type plasmids from *Gammaproteobacteria* [5,62,63]. A conserved *repAQ* tandem array is characteristic for these plasmids (Figure 2), and the respective genes of, e.g., the 6388 bp plasmid pHP18 from *Aeromonas caviae* HP18 were annotated as ‘helicase’ and ‘replication protein,’ respectively (NZ_KU644672.1). (ii) RepY: The putative replicase of pDSM110277_f showed a 68% sequence identity to the functional replication protein of the *Paracoccus marcusii* OS22 plasmid pMOS6 [25]. A naming according to the conserved Pfam protein domain ‘RepL’ from *Firmicutes* (PF05732) would be misleading due to the lack of any sequence homology with the recently described RepL-type plasmid replicase from *Rhodobacterales* [7]. Therefore, we named the novel replicase ‘RepY’ to indicate its unique evolutionary origin and distinguish the corresponding plasmids from all other plasmid types of *Rhodobacterales*. (iii) RepW: The putative replicase of the plasmids pDSM107133_l, pDSM109990_i, and pDSM110093_e was designated RepW. It showed homology to the Pfam protein family Rep_1 (PF01446) that initiates a rolling circle replication [64].

In accordance with other plasmids lacking a partitioning system, the new plasmid types are characterized by an increased copy number (Table 1). The highest number of 124 copies was observed for the RepY plasmid from *S. pontiacus* (pDSM110277_f), which is roughly equivalent to the copy number of cloning vectors in *E. coli* [4]. Moreover, this high copy number plasmid also has the lowest GC content of all investigated replicons (52.1%) and the largest difference to the respective chromosome (Δ8.4%). A comparable observation was made for the novel RepQ and RepW-type plasmids, whose GC content is at least 5% lower than those of the chromosome (Table 1). It is quite common that ‘intracellular genetic parasites’ such as cryptic plasmids, phages, and insertion sequences tend to be AT rich [65]. As G and C are less available and energetically more expensive, a reduced GC content makes these elements less expensive to carry for the host. Therefore, it might be selectively favored and may also promote higher copy numbers [40,65,66]. However, the observed difference may alternatively be explained by the horizontal acquisition of the RepQ, RepY, and RepW plasmids from phylogenetically distinct hosts with a lower GC content.

### 3.4. Characterization of RepQ, RepY and RepW-Type Plasmids

#### 3.4.1. Functionality of the Three Novel Plasmid Replicases

Plasmid replication of the putative replicases RepQ, RepY, and RepW were investigated in the model organism *Phaeobacter inhibens* DSM 17395 with our established functionality test [4,7]. We amplified the five newly discovered replication genes, including a 5′ upstream region of at least 450 bp and a 3′ downstream region of at least 100 bp, which should contain all essential cis-acting elements for plasmid replication, and cloned the PCR products into the *E. coli* vector pCR2.1. Successful transformation and replication in *P. inhibens* DSM 17395 verified the functionality of all five constructs (Appendix A) since the empty pCR2.1 vector generally does not replicate in *Alphaproteobacteria* (Petersen et al., 2019). The isolation of the constructs from *P. inhibens* and their subsequent successful re-transformation into *E. coli* provided the final proof of their autonomous replication in *Rhodobacterales*. Accordingly, we could document the functionality of the replicase RepQ from *S. indolifex* DSM 14862^T^, RepY from *S. pontiacus* DSM 110277, and three different RepW-type replicases from *S. dubius* DSM 109990, *Sulfitobacter* sp. DSM 110093, and *Pseudosulfitobacter* sp. DSM 107133.

#### 3.4.2. Gene Content of the Novel Plasmids

Despite its small size, the 17 kb RepQ plasmid pDSM14862_g of *S. indolifex* is obviously not cryptic. It encodes the quorum-sensing autoinducer 2 sensor kinase/phosphatase LuxQ (Figure 2, Appendix A), which regulates biofilm formation in *Vibrio* [67,68]. Beyond the conserved *repAQ* replication module, it contains several genes for plasmid-related functions such as the relaxase TraI, which is part of the MOB_P_ family, that likely mediates the mobilization of the replicon [69]. Another protein, which harbours a DNA-binding domain (Pfam: PF11740) and was annotated as chromosome partition protein Smc (HAMAP signature: MF_01894), might be involved in plasmid partitioning [70]. However, the calculated number of 21 plasmid copies per cell (Table 1) indicates that it has at most an auxiliary function for the RepQ plasmid [17]. The gene located downstream of *smc*, DSM14862_04460 (Figure 2, Appendix A), is homologous to the EcoRI methylase from *E. coli* (P00472; 52% aa identity) [71], which methylates (6-methyladenine) the second adenine of the palindromic sequence GAATTC. However, our PacBio sequencing data clearly documented that this motif is not methylated in *S. indolifex* DSM 14862^T^. Besides the 6-methyladenine modification motifs GANTC and RGATCY, which were detected in all *Sulfitobacter* genomes sequenced in the current study, the motif GGCATC was exclusively identified in strain DSM 14862^T^. This pattern has no match in ‘REBASE’ [72], so it remains to be investigated if its methylation is catalyzed by the gene product of DSM14862_04460. A corresponding restriction enzyme could not be unequivocally identified, but the adjacent gene (DSM14862_044601) contained an HNH endonuclease Pfam domain (PF01844.23), suggesting that the gene pair may represent a novel restriction-modification system.

The 7 kb RepY plasmid pDSM110277_f of *S. pontiacus* is a cryptic high copy number plasmid encoding only four genes beyond its eponymous replicase, i.e., a mobilization protein, a hypothetical protein, and a type II restriction-modification module (Figure 2). The respective methylation protein is annotated as ‘DNA (cytosine-5-)-methyltransferase’ (EC 2.1.1.37), but the corresponding modification motif has not yet been identified. The PacBio coverage (97x) was sufficient to detect two 6-methyladenine modification motifs mentioned above, but the detection of 5-methylcytosine requires a much higher sequencing depth (~250x). The adjacent gene encodes an NgoMIV family type II restriction enzyme [72], but the phylogenetic distance to well-characterized homologs allows no conclusion about the palindromic recognition site (EEZ47687: 56% aa identity). However, the presence of a comparable type II restriction-modification module on the cryptic plasmid pAES2 of *Paracoccus aestuarii* DSM 19484^T^ [25] is indicative of a functional role as a bacterial addiction module using post-segregational killing as a protection mechanism against plasmid loss [73,74].

The three RepW type replicons also represent small cryptic plasmids with a size of less than 6 kb, which carry only three to six annotated genes besides their replicase (Figure 2, Appendix A). Two of the RepW plasmids only contain hypothetical proteins without a known function, but the RepW plasmid of *Pseudosulfitobacter* sp. (pDSM107133_l) harbors a MOB gene indicating its transferability. pDSM107133_l also encodes the mRNA interferase toxin YafQ that inhibits protein translation and the corresponding antitoxin DinJ, whose mode of toxicity has been investigated in *E. coli* [75,76]. Plasmid encoded toxin-antitoxin modules represent another type of selfish genetic element that acts as an addiction module and ensures the maintenance of the mobile replicons [18]. The recent discovery of a plasmid-encoded toxin-antitoxin system that directly controls the plasmid copy number provides a new perspective on the functional role of these widespread modules for plasmid biology [77].

### 3.5. Distribution, Evolution and Function of the New Plasmid Types

#### 3.5.1. Presence of RepQ, RepY and RepW in Closed *Rhodobacterales* Genomes

The distribution of the major replicon types in our newly sequenced strains is representative of copiotrophic *Rhodobacterales* (Figure 1). In the set of 41 complete genomes, the RepABC type (86) was the most abundant, followed by RepA (34), DnaA-like (28), and RepB-type modules (24). These types were widely distributed among the different *Rhodobacterales* clades and even present in the distinct clade 8 comprising *inter alia* the genera *Paracoccus* and *Rhodobacter*. In contrast, RepC_soli, RepL, and RepW-type plasmids occurred only occasionally in the collection of 41 genome sequenced *Rhodobacterales* and RepQ, and RepY-type plasmids were only detected in a single *Sulfitobacter* strain. The sporadic and scattered distribution of the newly discovered RepQ, RepY, and RepW-type plasmids likely reflects their generally sparse occurrence within *Rhodobacterales*. However, as closed genomes only account for a small portion of genomes available at NCBI, we searched for all homologous replicases in the NCBI protein database. An extrachromosomal localization of several replicases could be confirmed, partially via manual circularization. Sequence sampling for phylogenetic maximum likelihood analyses, which was conducted with BLASTP searches, was aimed to identify (i) the closest homologs of the replicases (Figure 3, Figure 4 and Figure 5) and (ii) circular plasmids from more distantly related taxa (e.g., *Gammaproteobacteria*, Appendix A).

#### 3.5.2. RepQ-Type Plasmids

Comprehensive BLASTP searches revealed a very rare occurrence of RepQ type plasmids in *Rhodobacterales* (Figure 3). Apart from *S. indolifex* DSM 14862^T^, the replicase has only been identified in two *Roseovarius* sp. strains that were both isolated from the brown alga *Ectocarpus subulatus*. On the other hand, RepQ type plasmids are frequently found in different orders of *Gammaproteobacteria,* yet are missing in other alphaproteobacterial lineages. The closely related gammaproteobacterial sister lineage of the three *Rhodobacterales* replicases comprise small (5 to 10 kb) RepQ type plasmids, which were identified in *Oceanospirillales* (*Halomonas massiliensis* Marseille-P2426T), *Alteromonadales* (*Marinobacter antarcticus* CGMCC 1.10835), *Enterobacterales* (e. g., *Escherichia coli* LMLEEc034) and *Pseudomonadales* (*Pseudomonas cremoris* WS 5096; Figure 3, Appendix A). Like pDSM14862_g, all plasmids comprise mobilization genes as well as the characteristic *repAQ* cassette. The RepQ homologs sampled from *Betaproteobacteria* are always located on the chromosome (Appendix A), which might reflect an alternate function beyond plasmid replication in this bacterial class. However, we confirmed an extrachromosomal localization for the majority of RepQ proteins (20/34), and the size of the respective plasmids ranges from 5 to 53 kb. The small 5 kb replicons of *P. cremoris* and *Salmonella enterica* might represent cryptic plasmids, but virtually all other ECRs comprise additional genes.

The abundance of mobilization genes on RepQ type plasmids and the presence of identical replicases in different *Gammaproteobacteria* provide strong evidence that these replicons are potent mediators of horizontal gene transfer (HGT; Appendix A). A prime example for trans-order HGT is the nearly identical 7212 bp plasmids of *Aeromonas caviae* AB5 (*Aeromonadales*) and *Enterobacter roggenkampii* IPM1H6 (*Enterobacterales*) that differ in only one SNP. The presence of a *qnrS2* quinolone resistance gene, which was also identified on ten other ECRs analyzed in the current study (Appendix A), emphasizes the relevance of these plasmids for the horizontal spread of antibiotic resistance. The respective quinolone-resistant strains have been isolated from municipal wastewater treatment facilities in Germany and Israel [63,78] as well as rivers near hospitals and aquaculture in China [5] and other environments. The relevance of genetic recombination is exemplified for the 8789 bp plasmid pKA2Q of *Kluyvera* sp. KA2 that shares a nearly identical backbone of 6 kb with the plasmid pHP5 from *Aeromonas allosaccharophila* (5915/5927; 99.8% identity) but with the *qnrS2* gene replaced by a *bla_Fox_* gene conferring resistance against cephalosporin antibiotics. Some *qnrS2*-containing RepQ plasmids (*Escherichia coli* pUR19829-KPC21 and *Aeromonas taiwanensis* p1713-KPC) acquired an additional resistance gene against carbapenems (*bla_KPC-21_*), which are used as antibiotics of last resort [62].

The discovery of the functional RepQ type plasmid pDSM14862_g in *S. indolifex* DSM 14862^T^ expanded the host range of these promiscuous ECRs from *Gammaproteobacteria* to *Alphaproteobacteria*. The structural conservation of the plasmid backbone indicates that this replicon can be transferred between *Rhodobacterales*. Accordingly, it might mediate rapid environmental adaptations as previously shown for the small mobilizable RepL type chromate resistance plasmid of “*Candidatus* Marinibacterium anthonyi*”* DSM 107130 [7].

#### 3.5.3. RepY-Type Plasmids

The functionality of the novel replicase has been shown for *S. pontiacus* DSM 110277 (current study) and for *P. marcusii* OS22 [25], which represent a typical marine and non-marine *Rhodobacterales* strain, respectively. RepY proteins, which were present in eleven of about 3000 sequenced *Rhodobacterales* genomes, form a distinct alphaproteobacterial subtree with a basal positioning of *S. pontiacus* DSM 110277 (Figure 4). All other RepY proteins were found in non-marine taxa of clade 8 represented by the genera *Paracoccus*, *Rhodobacter,* and (*Falsi*-)*Gemmobacter* (Appendix A), which indicates that RepY-type plasmids are of minor relevance for the mobilome of the Roseobacter group. However, the discovery of the cryptic plasmid pDSM110277_f in the genus *Sulfitobacter* is notable due to its exceptional abundance (Table 1). With a calculated number of 124 copies per chromosome equivalent, it represents the first natural high copy number plasmid that has been discovered in roseobacters. Horizontal transfer of the alphaproteobacterial RepY-type plasmids is reflected by the nested positioning of *Bartonella* sp. HY038 (*Hyphomicrobiales*, *Bartonellaceae*), which probably recruited its 15 kb plasmid from the genus *Paracoccus*. This conclusion is supported by the presence of mobilization genes, e.g., on the plasmids of *Sulfitobacter* (Figure 2) and *P. marcusii* [25].

The alphaproteobacterial RepY replicons are generally small (6.5 to 15.0 kb). Those below 10 kb are supposed to be cryptic, but the phylogenetic sister group with beta- and gammaproteobacterial RepY proteins represents ECRs with sizes of up to 45 kb (Figure 4). Three plasmids of *Pseudomonas savastanoi* pv. *savastanoi* NCPPB 3335 were investigated for their role for tumor induction in olive plants [79]; the 42 kb RepY-type plasmid pPsv48C is not required for pathogenesis, but it encodes a putative isopentenyl-diphosphate delta-isomerase catalyzing a key step in isoprenoid biosynthesis. The essential gene *phzF* for the biosynthesis of phenazines, which represent heterocyclic nitrogen-containing metabolites with antibiotic and antitumor activity [80], was found on the RepY plasmid SGTM_pl1 of the betaproteobacterium *Sulfuriferula nivalis* SGTM. These extrachromosomal genes illustrate the relevance of RepY-type plasmids for *Proteobacteria*.

#### 3.5.4. RepW-Type Plasmids

The presence of three cryptic RepW-type plasmids in different (*Pseudo*)*sulfitobacter* strains, which were discovered in the current study, reflects a wider distribution among *Rhodobacterales* than RepQ- and RepY-type replicons (Figure 1). This novel plasmid type is mainly found in *Alpha*-, *Beta*-, and *Gammaproteobacteria,* and most alphaproteobacterial sequences form a distinct subtree (Figure 5 and Appendix A). In contrast to RepY, RepW replicases were detected in several clades of the marine Roseobacter group represented, e.g., by the genera *Ruegeria* (clade 1), *Sulfitobacter* (clade 2), *Sagittula* (clade 3), *Yoonia* (clade 4), and *Donghicola* (clade 8; Figure 5; Appendix A). The RepW proteins of the three (*Pseudo*)*sulfitobacter* plasmids are only distantly related, reflecting the diversity and horizontal exchange of RepW plasmids in *Rhodobacterales*.

We identified 25 circular RepW-type plasmids with a median size of only 2.6 kb (Appendix A). Although these plasmids are very small (2.2–6.8 kb), it is rather unlikely that they have been systematically overlooked in sequenced bacteria because most of the available genomes are based on Illumina shotgun assemblies. Nevertheless, the identification of twelve circular plasmids from the rat-gut metamobilome documents a great sampling gap of cryptic RepW-type plasmids (Figure 5, Appendix A). The sequences were determined from the cecal content of a dozen rats from Danish hospital sewers and the Falkland Islands [81], following up a seminal pilot study that was aimed to identify complete small plasmids in metagenome datasets [82]. Metamobilomics of a single animal intestine microbiome resulted in nearly the doubling (12/25) of the investigated circular(ized) RepW-type replicons, thus providing a first glimpse into the tremendous under-sampling of plasmids from natural habitats.

The characterization of the small cryptic RepW-type plasmids pTJ86-1 and pTJ86-2 from *Cuprividus taiwanensis* TJ86 (*Betaproteobacteria*, *Burkholderiales*; Appendix A, Appendix A) showed that these replicons are using a rolling-circle replication [83]. The derived shuttle vector pS4-tet^R^ was suitable for the transformation of the genera *Burkholderia* and *Cupriavidus*. Another example is the cryptic RepW plasmid pMGT of *Magnetospirillum magneticum* MGT-1 (*Alphaproteobacteria*, *Rhodospirillales*) that served as a backbone for the construction of the shuttle vector pUMG [84], which showed stability, a high copy number of about 40 plasmids per cell and allowed a higher gene expression of the tested reporter-luciferase construct. The novel 4511 bp RepW-type plasmid pDSM107133_l from *Pseudosulfitobacter* sp. DSM 107133 might be suitable for the development of a new cloning vector for *Rhodobacterales* because it already harbors a *mob* gene and a toxin-antitoxin system (Figure 2).

## 4. Conclusions

Current high throughput DNA sequencing technologies make it possible to decipher a complete bacterial genome for the price of a good bottle of wine, and they pave the way for the systematic characterization of ECRs. Based on the results of the present study, a combination of long-read and short-read sequencing is required to establish genomes of the highest quality encompassing all replicons and ideally lacking any sequencing errors. Conclusions about ECRs in the environment that were formerly drawn from a mobilization-dependent capturing of resistance plasmids from soil samples by conjugation into new host bacteria [3] or from metagenomic analyses of the gut mobilome [82] can now be complemented by holistic insights from an organismic perspective. A good starting point for understanding the organization and evolution of multipartite genomes is the differentiation between plasmids and chromids [40]. The classification of *Rhodobacterales* ECRs based on their crucial replicases allows to understand the individual genome architecture of a single strain, but it moreover provides first clues about commonalities and differences shared between related taxa. A prime example is the identification of a syntenic RepABC-8 chromid with a size of about 200 kb that is commonly shared by four closely related *Sulfitobacter* species (*S. dubius*, *Sulfitobacter* sp. DSM 110093, *S. profundi*, *S. indolifex*) but missing in the genomes of all more distantly related species of this genus. The acquisition of a new chromid is a unique and diagnostic event for the reconstruction of bacterial evolution. Its detection provides the promising perspective to identify key genes that once triggered bacterial speciation via the conquest of a novel ecological niche.

The central finding of the current study was the discovery of the three novel plasmid systems RepQ, RepY, and RepW based on a systematic assessment of all ECRs from five completely sequenced strains. The analyzed replicons are small cryptic, medium to high copy number plasmids without an obvious advantage for the bacterial host. However, they encode replication, stability, and mobility genes as plasmid backbone and could acquire beneficial genes that can easily be exchanged across species and genus borders [7,85]. These replicons might also promote evolutionary innovation as it was observed that multi-copy plasmids allow bacteria to escape from fitness trade-offs [86]. The wealth of different plasmid systems characterized in the current study also provides a promising perspective to develop new molecular tools for biotechnological applications in *Rhodobacterales*. Genetic complementation of roseobacters is usually performed with the medium copy number broad host range pBBR1MCS vector series [87,88], but neither low nor high copy number vectors are yet available. RepABC-derived vectors would be ideal for applications where a single gene copy per cell is required [89]. In contrast, the RepY-type plasmid from *S. pontiacus* DSM 110277 seems to be predestined to develop a high copy number vector for protein expression in *Rhodobacterales*.

## Figures and Tables

**Figure 1 microorganisms-10-00738-f001:**
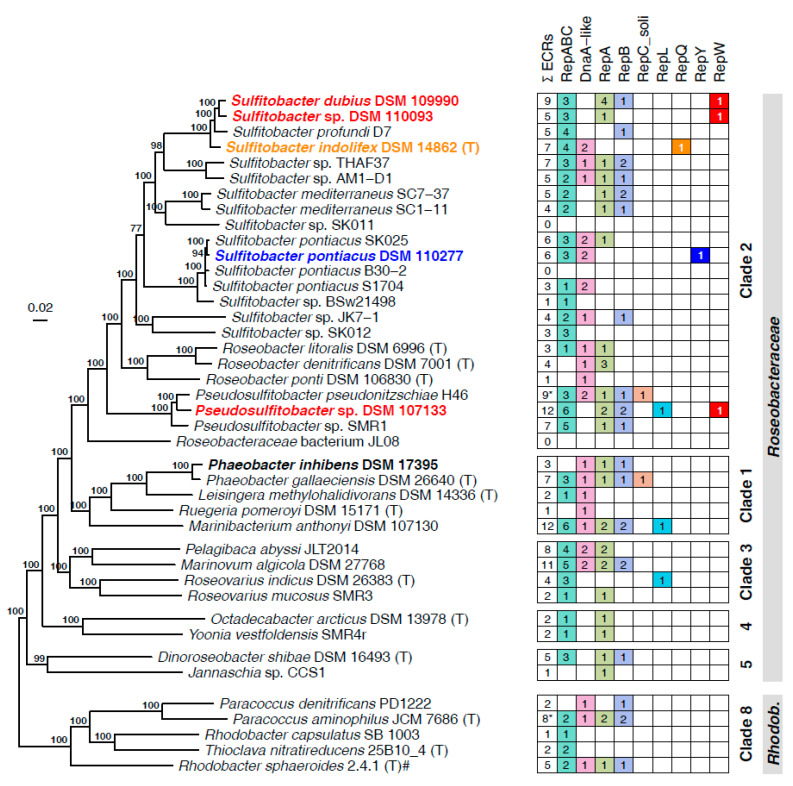
Phylogenomic tree of 41 *Rhodobacterales* strains with closed genomes and distribution of their extrachromosomal replicons (ECRs). The phylogenetic tree was constructed from 30,310 amino acid positions of 92 housekeeping genes. The monophyly of the family *Roseobacteraceae* and a sister group position of the *Rhodobacteraceae* (Clade 8) was confirmed by a phylogenomic analysis with the draft genome of the deep branching taxon *Neomegalonema perideroedes* DSM 15528. The matrix on the right side depicts the number of ECRs classified by their replication system. An asterisk (*) indicates that one of the ECRs contained only a putative replicase, which function has not been tested in *Rhodobacterales*. Strains sequenced and investigated within this study are colored in accordance with their novel plasmid systems and shown in bold. # The correct taxonomic name is *Cereibacter sphaeroides* 2.4.1. Genome accession numbers are listed in Appendix A.

**Figure 2 microorganisms-10-00738-f002:**
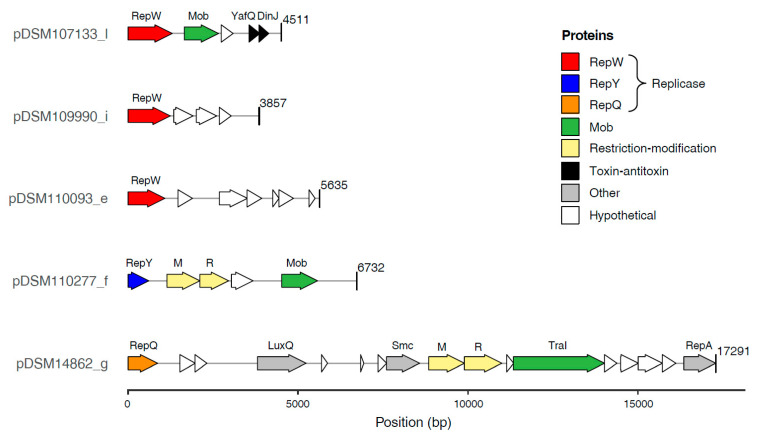
Gene content of five (*Pseudo*)*sulfitobacter* plasmids representing the novel replicon types RepW, RepY and RepQ. The plasmids were identified in the complete genomes from *Pseudosulfitobacter* sp. DSM 107133, *S. dubius* DSM 109990, *Sulfitobacter* sp. DSM 110093, *S. pontiacus* DSM 110277 and *S. indolifex* DSM 14862^T^ (Table 1). The figure was generated with the R package gggene, respective annotations are shown in Appendix A.

**Figure 3 microorganisms-10-00738-f003:**
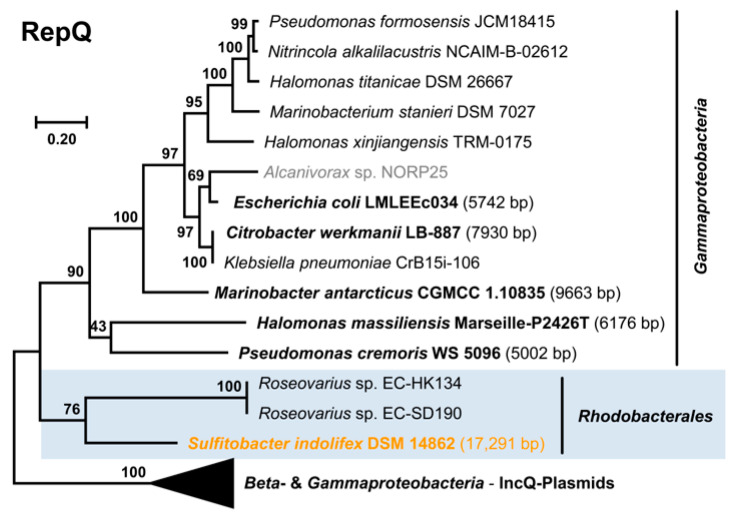
Phylogenetic Maximum Likelihood tree of 34 RepQ type plasmid replication proteins based on 266 amino acid positions. The replicase of *Sulfitobacter indolifex* DSM 14862^T^ is shown in orange, and replicases from metagenome-assembled genomes (MAGs) are shown in gray. Circular plasmids or circularized contigs are highlighted in bold. Sequences from *Rhodobacterales* are highlighted by a blue box. The complete tree is shown in Appendix A. Accession numbers, localization, and further characteristics of RepQ type plasmids are presented in Appendix A.

**Figure 4 microorganisms-10-00738-f004:**
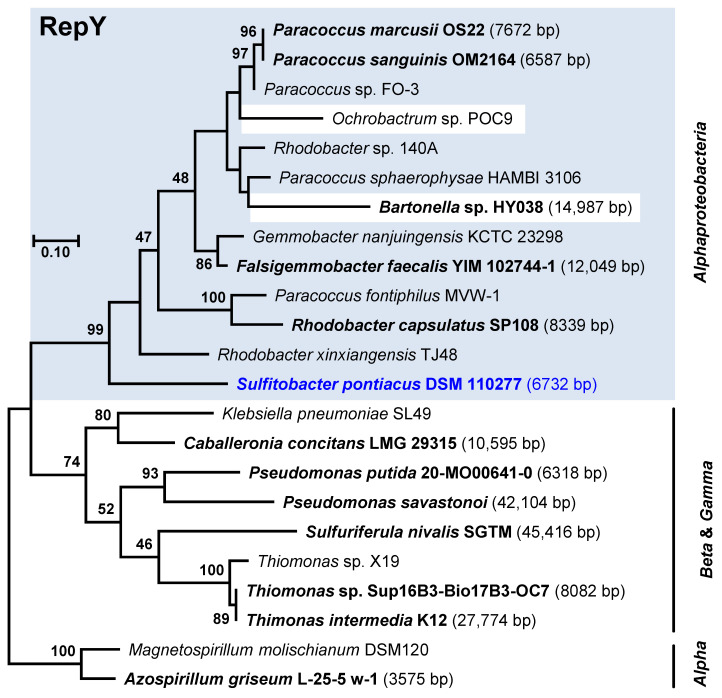
Phylogenetic Maximum Likelihood tree of 23 RepY type plasmid replication proteins based on 157 amino acid positions. Bootstrap support > 40% is indicated. The replicase of *Sulfitobacter pontiacus* DSM 110277 is shown in blue. Circular plasmids or circularized contigs are highlighted in bold. Sequences from *Rhodobacterales* are highlighted by a blue box. Accession numbers, localization, and further characteristics of RepY type replicases are presented in Appendix A.

**Figure 5 microorganisms-10-00738-f005:**
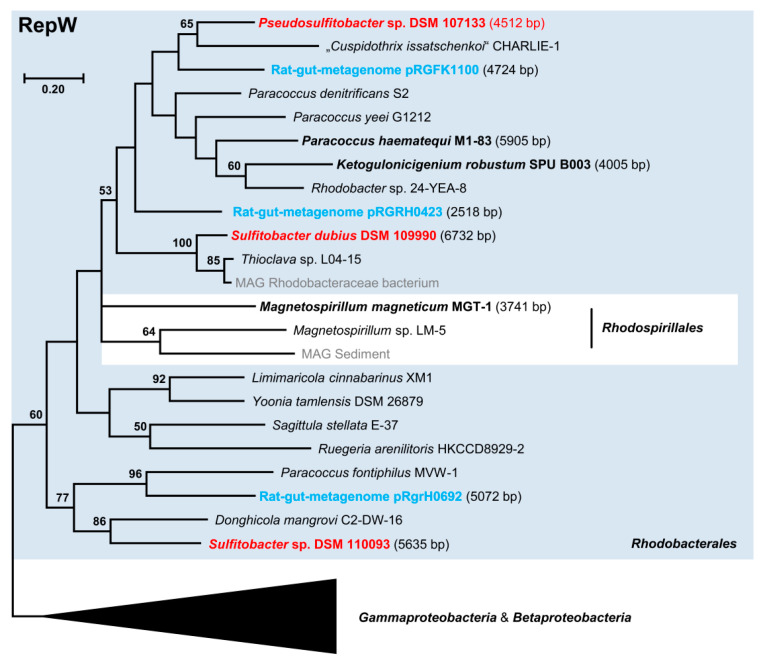
Phylogenetic Maximum Likelihood tree of 61 RepW type plasmid replication proteins based on 123 amino acid positions. Bootstrap support >40% is indicated. The reference replicases of *Pseudosulfitobacter* sp. DSM 107133, *Sulfitobacter dubius* DSM 109990, and *Sulfitobacter* sp. DSM 110093 are shown in red. Circular plasmids or circularized contigs are highlighted in bold. Rat-gut metamobilome-derived sequences are shown in blue, and sequences from MAGs are shown in gray. Sequences from *Rhodobacterales* are highlighted by a blue box. The complete tree is shown in Appendix A. Accession numbers, localization, and further characteristics of RepW type replicases are presented in Appendix A.

**Table 1 microorganisms-10-00738-t001:** Replicon characteristics of investigated (*Pseudo*)*sulfitobacter* strains with closed genomes.

Replicon Id	Size (bp)	Replicon Type	GC	Copy No.	Replication Module	Partitioning System	Mobility	NCBI Accession
** *Sulfitobacter indolifex* ** **DSM 14862 (HEL-45)**								
cDSM14862	3,271,523	chromosome	60.0	1.0	DnaA	yes	no	CP084951
pDSM14862_a	313,826	plasmid	55.4	0.5	RepABC-20	yes	no	CP084952
pDSM14862_b	307,297	plasmid	56.1	0.8	DnaA-like-I	yes	no	CP084953
pDSM14862_c	200,719	chromid	60.3	0.8	RepABC-8	yes	no	CP084954
pDSM14862_d	160,516	plasmid	56.5	0.8	DnaA-like-II	yes	no	CP084955
pDSM14862_e	123,234	plasmid	57.1	0.6	RepABC-4	yes	T4SS	CP084956
pDSM14862_f	104,164	plasmid	59.8	0.7	RepABC-7	yes	T4SS	CP084957
pDSM14862_g	17,291	plasmid	54.9	5.3	RepQ	no	MOBP	CP084958
** *Sulfitobacter pontiacus* ** **DSM 110277 (SO248Ex84)**								
cDSM110277	3,012,962	chromosome	60.5	1.0	DnaA	yes	no	CP084959
pDSM110277_a	239,416	plasmid	60.2	0.7	RepABC-10	yes	T4SS	CP084960
pDSM110277_b	230,345	chromid	59.7	0.7	DnaA-like-II	yes	no	CP084961
pDSM110277_c	177,750	chromid	60.3	0.6	DnaA-like-I	yes	no	CP084962
pDSM110277_d	128,813	plasmid	59.6	0.5	RepABC-9-1	yes	T4SS	CP084963
pDSM110277_e	53,945	plasmid	57.0	1.8	RepABC-8	yes	no	CP084964
pDSM110277_f	6732	plasmid	52.1	124.3	RepY	no	MOBQ	CP084965
***Sulfitobacter* sp. DSM 110093 (2RS2_G6)**								
cDSM110093	3,434,207	chromosome	59.7	1.0	DnaA	yes	MOBV	CP085167
pDSM110093_a	336,544	plasmid	55.3	1.1	RepABC-21	yes	no	CP085168
pDSM110093_b	273,772	plasmid	54.9	1.1	RepABC-20	yes	no	CP085169
pDSM110093_c	268,356	chromid	59.7	1.0	RepABC-8	yes	no	CP085170
pDSM110093_d	21,520	plasmid	52.6	4.3	RepA	no	MOBP	CP085171
pDSM110093_e	5635	plasmid	53.9	23.4	RepW	no	no	CP085172
***Sulfitobacter dubius* DSM 109990 (3RS2_G4b)**								
cDSM109990	3,274,709	chromosome	60.2	1.0	DnaA	yes	no	CP085144
pDSM109990_a	284,454	plasmid	55.4	1.2	RepB-I	yes	MOBP	CP085145
pDSM109990_b	247,035	chromid	60.5	0.9	RepABC-8	yes	no	CP085146
pDSM109990_c	183,486	plasmid	57.6	0.9	RepABC-2	yes	no	CP085147
pDSM109990_d	108,277	plasmid	59.9	0.6	RepABC-1	yes	T4SS	CP085148
pDSM109990_e	47,721	plasmid	58.7	2.6	RepA_a	no	MOBQ	CP085149
pDSM109990_f	6286	plasmid	59.7	9.5	RepA_b	no	MOBQ	CP085150
pDSM109990_g	4878	plasmid	52.3	12.5	RepA_c	no	no	CP085151
pDSM109990_h	4609	plasmid	58.4	12.4	RepA_d	no	MOBV	CP085152
pDSM109990_i	3857	plasmid	55.0	15.3	RepW	no	no	CP085153
***Pseudosulfitobacter* sp. DSM 107133 (PIC-76)**								
cDSM107133	3,635,847	chromosome	61.1	1.0	DnaA	yes	no	CP085154
pDSM107133_a	571,401	chromid	61.0	1.1	RepABC-3	yes	no	CP085155
pDSM107133_b	246,683	chromid	61.1	0.6	RepB-I	yes	no	CP085156
pDSM107133_c	158,961	plasmid	60.3	0.5	RepABC-11	yes	T4SS	CP085157
pDSM107133_d	135,500	plasmid	58.0	0.4	RepABC-9-1	yes	MOBP	CP085158
pDSM107133_e	129,047	plasmid	58.4	0.4	RepABC-5	yes	T4SS	CP085159
pDSM107133_f	128,511	plasmid	60.2	0.4	RepABC-1	yes	T4SS	CP085160
pDSM107133_g	105,694	plasmid	60.4	0.4	RepABC-9-2	yes	T4SS	CP085161
pDSM107133_h	65,426	chromid	63.6	0.6	RepB-III	yes	no	CP085162
pDSM107133_i	17,449	plasmid	53.1	3.8	RepA_a	no	no	CP085163
pDSM107133_j	7190	plasmid	59.6	9.1	RepA_b	no	MOBQ	CP085164
pDSM107133_k	5670	plasmid	59.0	8.6	RepL	no	MOBQ	CP085165
pDSM107133_l	4511	plasmid	53.5	21.3	RepW	no	MOBV	CP085166

## Data Availability

Genome sequences are deposited in NCBI GenBank under the accession numbers: CP084951-58, CP084959-65, CP085144-53, CP085154-66 and CP085167-72.

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
