# Peer review of "Beyond the ABCs—Discovery of Three New Plasmid Types in Rhodobacterales (RepQ, RepY, RepW)"

_microorganisms, 2022, doi:10.3390/microorganisms10040738_

Round 1

Reviewer 1 Report

The paper by Petersen et al.  reports complete genomes of Roseobacteraceae taxa related to Sulfitobacter and the discovery of three novel plasmids in such genomes. The paper is overall well written and based on sound genomic analysis.

However, the paper lacks depth and phylogenomic perspective regarding the order Rhodobacterales, missing key references on the topic, in particular:

Liang, K. Y., Orata, F. D., Boucher, Y. F., & Case, R. J. (2021). Roseobacters in a sea of poly-and paraphyly: whole genome-based taxonomy of the family Rhodobacteraceae and the proposal for the split of the “Roseobacter clade” into a novel family, Roseobacteraceae fam. nov. Frontiers in Microbiology, 12, 1635.

In this and other recent papers, for instance Hordt et al (2020) - se below - Sulfitobacter is reported to be poly-phyletic. The phylogenomic-based taxonomic analysis on the new genomes described in the paper is inadequate to evaluate the exact phylogenetic position of the new taxa related to this poly-phyletic genus.

While the Authors describe in the Methods the phylogenetic inference, model and program for the ML trees shown for the Rep proteins in Figs. 3-5, they do no provide an adequate set of  information for the critical phylogenomics analysis of Fig. They cite a software package for tree rendering (line 170) and  Ref. [43], which has been originally applied to E.coli, not to Alphaproteobacteria or general bacterial taxonomy. The Authors should compare the phylogeny they report with that obtained with the popular and universal  GTDB. Moreover, the tree shown in Fig. 1 has limited taxonomic sampling, since it misses deep branching Rhodobacterales such as Neomegalonema (see Hordt et al, 2020, Frontiers in Microbiol), which are critical for rooting the internal clades.

Another comment regards the taxonomic distribution of the new small plasmids reported in the paper. The finding that the such plasmids of Rhoseobacteraceae are predominantly distributed in marine Gammaproteobacteria (Section 3.5.2) is interesting for their relevance of LGT. A classical example of LGT from gammaproteobacteria to Rhodobacter is the second flagellum operon. How many operons for flagella are present in the new genomes of Rhodobacterales described ?

In sum, the paper requires extensive revision to widen its scientific relevance and refine the phylogenomics  of the taxa reported.

Author Response

Comments to the Author

Comments and Suggestions for Authors

The paper by Petersen et al.  reports complete genomes of Roseobacteraceae taxa related to Sulfitobacter and the discovery of three novel plasmids in such genomes. The paper is overall well written and based on sound genomic analysis.

However, the paper lacks depth and phylogenomic perspective regarding the order Rhodobacterales, missing key references on the topic, in particular:

Liang, K. Y., Orata, F. D., Boucher, Y. F., & Case, R. J. (2021). Roseobacters in a sea of poly-and paraphyly: whole genome-based taxonomy of the family Rhodobacteraceae and the proposal for the split of the “Roseobacter clade” into a novel family, Roseobacteraceae fam. nov. Frontiers in Microbiology, 12, 1635.

--- The aim of our current paper was the description of three novel plasmid types and not a comprehensive phylogenomic re-assessment of the order Rhodobacterales. Our taxon sampling for Figure 1 is based on closed reference genomes of our five newly sequenced Sulfitobacter and Pseudosulfitobacter strains of the Rhodobacterales Clade 2 (Newton et al. 2010 [ISME J]), because draft genomes were not sufficient for our comparative ECR analyses. Furthermore, we included some reference genomes of Clade 1, 3, 4, 5 and 8 representing the phylogenetic width of Rhodobacterales (Bartling et al. 2018 [Systematic and Applied Microbiology], Petersen et al. 2019 [PNAS]). The correct and meaningful rooting of our tree is illustrated below. We are grateful for referring to the publication of Liang et al. 2021 [Frontiers in Microbiology], which proposed to classify the marine roseobacters (= Figure 1, Clade 1 – 5) as Roseobacteraceae and to split this novel family from Rhodobacteraceae that comprise the non-marine genera of Clade 8.  Moreover, based on their phylogenomic analysis Liang et al. proposed the genus Pseudosulfitobacter for the wrongly classified species Sulfitobacter pseudonitzschiae. We now quote this important publication in our manuscript and included the deep split between Roseobacteraceae and Rhodobacteraceae in Figure 1.

In this and other recent papers, for instance Hordt et al (2020) - se below - Sulfitobacter is reported to be poly-phyletic. The phylogenomic-based taxonomic analysis on the new genomes described in the paper is inadequate to evaluate the exact phylogenetic position of the new taxa related to this poly-phyletic genus.

--- As the paraphyletic state of the genus Sulfitobacter in Hördt et al. (2020) was solely caused by the species S. pseudonitzschiae, this taxonomic incongruence was resolved by the proposal of the new genus Pseudosulfitobacter by Liang et al. (2021). However, strain JL08, which is designated Sulfitobacter sp. in the NCBI database, did not belong to either genus ([i] JL08 versus Sulfitobacter pontiacus DSM 10014T, AAI: 71.3; [ii] JL08 versus Pseudosulfitobacter pseudonitzschiae DSM 16824T, AAI: 71.4), therefore we originally presented the name in quotation marks in Figure 1. To avoid misunderstandings we now named the strain “Roseobacteraceae bacterium JL08”.

--- We would like to clarify that the classification of our new strains was based on a phylogenomic analysis that contained all type stains of the genera Sulfitobacter, Pseudosulfitobacter and Roseobacter (Figure S2; see also line 225-227). Accordingly, our proceeding was absolutely adequate for the taxonomic assessment of the investigated strains. Furthermore, our genomic analysis indicated that the species Sulfitobacter sabulilitoris (HSMS−29) and the proposed, but not validly published species ”Sulfitobacter algicola” (1151; see: https://lpsn.dsmz.de/species/sulfitobacter-algicola) were also wrongly assigned to the genus Sulfitobacter and should thus be reclassified. Nevertheless, these taxa had no influence on the phylogenetic positioning and the correct taxonomic assignment of our strains, because the affiliation to a genus is generally defined by the type species of a genus and our strains are located in two maximally supported monophyletic clusters containing the type species Sulfitobacter pontiacus DSM 10014T and Pseudosulfitobacter pseudonitzschiae DSM 16824T, respectively. For clarification, in lines 225-227 we now say “An additional well supported phylogenomic tree with all related type strains allowed us to identify the closest relative(s) of our four new strains (Figure S2)”.

While the Authors describe in the Methods the phylogenetic inference, model and program for the ML trees shown for the Rep proteins in Figs. 3-5, they do no provide an adequate set of  information for the critical phylogenomics analysis of Fig.

--- As we used the default model, program and parameters of the cited UBCG method (Na et al. 2018 [Journal of Microbiology]), we originally did not elaborate on the details. However, in the Methods part we now added the program and explicitly stated that we used the default parameters.  It now says: “A genome-based phylogenetic tree was reconstructed from a concatenated amino acid alignment of 92 housekeeping genes, which were generated with UBCG [44]. An approximately Maximum Likelihood phylogenetic tree was inferred using FastTree version 2.1.11 (double-precision) with default parameters [45]”.

They cite a software package for tree rendering (line 170) and  Ref. [43], which has been originally applied to E.coli, not to Alphaproteobacteria or general bacterial taxonomy. The Authors should compare the phylogeny they report with that obtained with the popular and universal  GTDB.

--- UBCG is also a popular and universal method for the reconstruction of phylogenomic trees with 478 (UBCG) versus 567 (GTDB) citations in the Web of Science. Furthermore, UBCG was applied for a wide range of taxa including Rhodobacterales (Alphaproteobacteria) in 260 IJSEM articles, which is the ultimate journal for taxonomic classification, while GTDB is commonly used for metagenome analyses. For UBCG as well as GTDB, housekeeping genes are extracted from the genomes using prodigal and subsequently identified with HMMER in comparison to Pfam and Tigerfam. UBCG represents a total set of 28 phyla, and not just E. coli. The database of UBCG is smaller than GTDB, because it uses a single complete genome per species, but no (potentially contaminated) metagenome assemblies as GTDB. Furthermore, UBCG restricted the set of housekeeping genes to the ones that occur with a single copy in at least 95% of the species, which should lead to a very stable backbone.

--- For the correct classification of the strains, we compared them with species with validly published names, which can be found in the List of Prokaryotic names with Standing in Nomenclature (LPSN) and in the validation lists of the International Journal of Systematic and Evolutionary Microbiology (IJSEM). Although, GTDB also uses LPSN as primary nomenclatural reference, the latest release was in June 2021 and for, instance, the genus Pseudosulfitobacter, which was proposed in Liang et al. 2021, was just validated at the beginning of 2022 (https://www.microbiologyresearch.org/content/journal/ijsem/10.1099/ijsem.0.005167;jsessionid=rn4kGeKws7_cxa38mZ696Ufj.mbslive-10-240-10-125) and thus not yet included in GTDB. Furthermore, valid published Sulfitobacter species names like S. profundi and S. maritimus are also lacking.

--- Finally, we also applied the GTDB toolkit to our dataset to generate an untrimmed alignment of the user genomes, which contained 41,083 amino acid positions retrieved from 119 genes. The topology of the resultant phylogenomic tree (see Figure A, below) had an absolutely identical topology to the tree in Figure 1. Slight differences in the branch length of the UBCG and GTDB tree might depend on the missing filtering step of gap positions in the GTDB alignment. Accordingly, we clearly showed that the phylogenomic tree in Figure 1 is absolutely sound.

Moreover, the tree shown in Fig. 1 has limited taxonomic sampling, since it misses deep branching Rhodobacterales such as Neomegalonema (see Hordt et al, 2020, Frontiers in Microbiol), which are critical for rooting the internal clades.

--- For the analysis of extrachromosomal replicons complete closed genomes were required, which severely restricted our taxon sampling. However, the maximal statistical support for the monophyly of the formerly identified distinct lineages of Rhodobacterales (Clade 1 to 8; Petersen et al. 2019 [PNAS]) provide clear evidence for a reliable phylogenomic reconstruction. The tree was correctly rooted with Clade 8 according to our former analyses with more distant outgroup sequences (Bartling et al. 2018 [Systematic and Applied Microbiology]). Nevertheless, we confirmed the topology that is presented in Figure 1 by adding the draft genome of the deep branching taxon Neomegalonema perideroedes DSM 15528 into our analysis (see Figure B, below). There was neither a difference in the topology of Roseobacteraceae nor in those of Rhodobacteraceae, and the phylogenomic tree also showed the monophyly of both families. According to the result of our new analysis, we added the following sentence to the legend of Figure 1: “The monophyly of the family Roseobacteraceae and a sister group position of the Rhodobacteraceae (Clade 8) was confirmed by a phylogenomic analysis with the draft genome of the deep branching taxon Neomegalonema perideroedes DSM 15528 (data not shown)”.

Another comment regards the taxonomic distribution of the new small plasmids reported in the paper. The finding that the such plasmids of Rhoseobacteraceae are predominantly distributed in marine Gammaproteobacteria (Section 3.5.2) is interesting for their relevance of LGT. A classical example of LGT from gammaproteobacteria to Rhodobacter is the second flagellum operon. How many operons for flagella are present in the new genomes of Rhodobacterales described ?

--- We agree that the discovery of three different and occasionally plasmid-located flagella operons in Rhodobacterales (Frank et al. 2015 [Environmental Microbiology]) exemplifies the relevance of HGT  and ECRs for the evolution of this bacterial order.  We investigated the distribution of the flagella systems in our five newly sequenced genomes, but all of them contained just a single operon.

In sum, the paper requires extensive revision to widen its scientific relevance and refine the phylogenomics  of the taxa reported.

--- Our state-of-the-art bioinformatic analyses provided robust and reliable results. Accordingly, we are convinced that all of our conclusions drawn from phylogenomic analyses are justified. We hope that our detailed response clarified the open questions and misunderstandings.

Reviewer 2 Report

Authors herein presented a comparative genomic survey in 5 bacteria of the roseobacter group leading to the definition of the existence of 3 new plasmids. Authors reported either pacbio and illuminate sequencing-based strategies followed by comparative analyses. The experimental section is well organised and contains the required details, similarly the results are well proposed and supported the conclusion of the authors. As a note, the authors also included the amplicon size. This was well appreciated.

Author Response

Authors herein presented a comparative genomic survey in 5 bacteria of the roseobacter group leading to the definition of the existence of 3 new plasmids. Authors reported either pacbio and illuminate sequencing-based strategies followed by comparative analyses. The experimental section is well organised and contains the required details, similarly the results are well proposed and supported the conclusion of the authors. As a note, the authors also included the amplicon size. This was well appreciated.

--- Thank you for the positive evaluation of our work and the kind feedback.

Round 2

Reviewer 1 Report

The detailed response has been appreciated.